# Understanding the Underlying Psychosocial Determinants of Safe Food Handling among Consumers to Mitigate the Transmission Risk of Antimicrobial-Resistant Bacteria

**DOI:** 10.3390/ijerph17072546

**Published:** 2020-04-08

**Authors:** Claudia Freivogel, Vivianne H. M. Visschers

**Affiliations:** School of Applied Psychology, University of Applied Sciences and Arts Northwestern Switzerland, 4600 Olten, Switzerland; vivianne.visschers@fhnw.ch

**Keywords:** antimicrobial resistance, consumer behavior, health action process approach, psychosocial determinants, safe food handling, subjective norm

## Abstract

In this study, we aimed to explore the psychosocial factors related to consumers’ safe food-handling behaviors to reduce the transmission risk of antimicrobial-resistant bacteria through food products. To this end, we investigated the extent to which the variables of the health action process approach (HAPA) and subjective norms can predict safe food handling by consumers. Data were collected from a representative sample of consumers belonging to the German-speaking part of Switzerland by administering a self-reporting questionnaire. The results showed that risk perception, self-efficacy, and positive outcome expectancy (i.e., the motivational phase of HAPA) were significant predictors of the intentions to handle food in a safe manner (see graphical abstract below). Additionally, in the volitional phase of HAPA, intention was found to be a significant predictor of safe food-handling behaviors. Contrary to expectations based on HAPA, action planning did not mediate the intention–behavior relationship. Only a small part of the variance in behavior was explained by coping planning and action control. The findings confirmed that the motivational phase of HAPA may be useful for determining safe food handling.

## 1. Introduction

Antimicrobial resistance (AMR; see acronyms in the Appendix A, Table A1) has become a global threat to public health [1], and it has devastating economic and health impacts on human society as the occurrence frequency and scope of the diseases increase [2,3]. The use of antimicrobials invariably leads to the selection of bacteria that are resistant to the substance used [4]. Resistant bacteria can spread through multiple pathways to different reservoirs, such as soil and animals [5,6,7]. Animal food products constitute an important pathway for the exposure of AMR bacteria to humans [8,9]. Therefore, food preparers can reduce the spread of AMR by engaging in preventive behavior, namely, adopting safe food-handling measures. It is essential to identify which psychosocial factors predict safe food-handling behaviors among food preparers to foster the implementation of such preventive measures. Thus, this study aimed to identify and quantify the relevant determinants of safe food-handling behaviors among food preparers, to reduce the transmission risk of AMR.

### 1.1. AMR in Animal Food Products

The use of antimicrobials in food-producing animals contributes to the development of AMR, especially in settings involving intensive animal production [10]. AMR bacteria can contaminate our food during slaughtering and processing [8]. Once food contaminated with AMR bacteria enters household kitchens, the resistant bacteria can reach consumers through cross-contamination and recontamination events [11]. In particular, chicken meat is a major source of AMR bacteria [12]. In samples collected by the National Antibiotic Resistance Monitoring program in Switzerland, 38.8% of domestic poultry meat and 57.3% of foreign poultry meat were found to be contaminated with Campylobacter jejuni or Campylobacter coli [13]. Campylobacter causes gastrointestinal infections, and it has been found to be resistant to commonly-used antibiotics [14,15].

Unhygienic food handling at home accelerates the spread of AMR [1]. More cases of foodborne illness are caused by food prepared at home than by food consumed in other settings [16,17]. Thus, AMR bacteria are transmitted in the same way as other foodborne germs. Food-handling behaviors in private households influence the transmission of AMR bacteria. Adopting basic hygiene is an effective preventive measure for avoiding contamination with AMR bacteria [18,19]. The spread of bacteria in the kitchen environment and to prepared meals can mainly be explained by a lack of thorough washing of contaminated hands, knives, and chopping boards both during and after meal preparation [20,21]. In addition to cross-contamination, the consumption of undercooked meat and poultry is another factor responsible for foodborne infections [22].

The Swiss Federal Food Safety and Veterinary Office [23] recommends four food-safety principles to reduce the risk of food poisoning caused by, for example, AMR bacteria: (1) cleaning of hands and kitchen utensils directly after contact with raw meat; (2) cooking meat thoroughly; (3) separating raw meat from ready-to-eat food; and (4) chilling raw meat to safe temperatures. Not all of these measures are known to consumers, let alone implemented regularly [24,25].

### 1.2. Psychological Determinants of Safe Food Preparation

Several behavioral theories explain which psychosocial constructs predict preventive health behavior. The factors examined in this study are based on the health action process approach (HAPA) [26] and the theory of planned behavior (TPB) [27]. Both models are clearly specified and have been effective in predicting hygiene-related behaviors, such as hand hygiene or safe food-handling practices [28,29,30]. The two theories have been integrated in several studies because the processes they explain can be complementary, e.g., [31,32]. That is, the TPB explains the determinants of intention of behavior, while HAPA additionally explains the mediators of the intention–behavior relationship.

According to the TPB, behavioral intention is shaped by subjective norms, perceived behavioral control, and attitudes [27,33]. The intention to demonstrate a given behavior is the most salient predictor of future behavior. However, findings have indicated that the model is better at predicting intention than behavior [29]. The TPB does not explain the intention–behavior gap, according to which health behaviors are not always performed in a manner that is consistent with one’s intentions, probably because other factors directly influence behavior [34,35].

The HAPA was developed to bridge the intention–behavior gap and therefore included post-intentional planning, which yielded a motivational phase and a volitional phase [26,36]. The motivational phase includes three predictors of intention (see Figure 1). The first predictor, risk perception, is a combination of subjective beliefs related to the likelihood of occurrence of a danger and the potential severity of the danger. The second predictor is outcome expectancies, which pertains to beliefs held by individuals about the expected positive (pros) and negative (cons) consequences of adopting or not adopting a behavior. The third predictor, self-efficacy, is related to beliefs about one’s abilities to perform a certain behavior, even when confronted with potential barriers.

The motivational factors in HAPA are expected to have stronger effects on intention than on behavior. They were found to predict the intention for safe food handling, whereby beliefs about the probability of contracting a risk constituted the strongest predictor [37]. Compared to perceived probability of risk, beliefs about the seriousness of a possible harm were not found to be related to intention. However, in a study on poultry preparation, perceived severity of foodborne illnesses was strongly correlated to safe food-handling behaviors [38]. Furthermore, a systematic review demonstrated that risk perceptions were positively related to several behavioral outcomes of safe food handling [39].

According to the HAPA, once an individual develops an intention, the volitional phase is entered [36,40]. In this phase, skills and strategies are considered to play major roles in the pursuit of the behavioral goal [40]. Intentions are more likely to translate into behaviors when people plan to attain a concrete behavioral goal and prepare to overcome the barriers hindering its achievement [41]. Therefore, research has incorporated two types of planning, as presented in Figure 1, e.g., [42]. Action planning pertains thoughts of when, where, and how one intends to perform a behavior. Coping planning involves developing strategies for dealing with potential anticipated barriers. Several studies have specified action planning and coping planning as mediators between intention and action, e.g., [43,44]. Action control as an additional self-regulatory determinant in the volitional phase has found empirical support in the literature [40,45]. It is an ongoing process in which one’s own behavior is continuously evaluated with regards to a behavioral standard.

The mediators provided by the HAPA may explain a part of the intention–behavior gap related to safe food handling. Action planning, coping planning, and action control were found to be important predictors of hand hygiene [46], which is a part of safe food handling. However, in a study examining the volitional phase of hygienic food handling, action planning did not mediate the effect between intention and behavior [29]. The authors argued that action planning may not have been important because participants believed that they usually prepare food hygienically. The study addressed hygienic food handling in general instead of specific food-handling measures (e.g., using different kitchen utensils for raw meat), which may explain why the participants were optimistic about their safe food-handling behavior. Generally, the HAPA factors have been successful in predicting intention to engage in various health behaviors, for instance, sunscreen usage and healthy eating [47,48,49]. Furthermore, interventions tackling the relevant HAPA determinants enhanced preventive health-related behaviors, e.g., [49].

However, the HAPA model does not consider possible social influences, such as subjective norms. According to the TPB, subjective norms predict intention and describe the perceived expectations from others (e.g., family and friends) to demonstrate a given behavior. Subjective norms have been found to be significant predictors of safe food-handling behaviors [28,30]. A study by Chow et al. [37] included a subjective norm component in the HAPA and found that the included component significantly improved the proportion of variance explained in the intention to implement safe food-handling behaviors. A systematic review showed that food preparers’ subjective norms and habits exhibited the most consistent relationships with behavioral intentions and behaviors across various safe food-handling measures [39]. Furthermore, food preparers were more likely to use food thermometers and wash hands adequately during food preparation when they perceived stronger subjective norms associated with these behaviors [50].

Moreover, a connection between past experiences and future behavior has been found for different behaviors, such as food handling or travel behavior [51,52]. This relationship is stronger when previous risk experience is associated with involuntary risk activities compared to voluntary activities [53]. A lack of negative experiences with foodborne pathogens has been found to foster loose food handling [54]. Vice versa, people who had suffered from a Campylobacteriosis reported more favorable food-handling behaviors than people who had not suffered from it [55]. According to the availability heuristic, the perceived likelihood of a risk increases if it has been experienced or can be readily imagined [56]. Therefore, seemingly, experience is used as a filter to evaluate and prioritize the risks involved in daily life [57].

### 1.3. Aims and Hypotheses

In sum, consumers’ perceptions and behaviors related to safe food handling to prevent the spread of AMR bacteria have not been investigated in detail thus far. Moreover, although the HAPA model has been employed in studies related to hygienic food handling, it has never been empirically tested in the context of safe food handling to reduce the transmission risk of AMR bacteria. Because the HAPA model is an open framework, the number and types of constructs in the model may vary according to the health behavior examined in a given study [40]. Identification of the factors underlying safe food-handling behaviors against the transmission risk of AMR bacteria is useful for developing strategies to reduce the spread of those resistant bacteria.

With this background, this study primarily aimed to identify the theoretical constructs of the HAPA model plus subjective norms that may be used to develop an intervention for mitigating the transmission risk of AMR bacteria through food. The secondary aim was to investigate the influence of past negative experiences on safe food-handling behaviors. Personal negative experiences with AMR bacteria might affect risk perception, which, in turn, might influence the likelihood of implementation of safe food-handling behaviors in the future to reduce the transmission risk of AMR bacteria.

The proposed research model based on the HAPA framework is illustrated in Figure 1. It is expected that intentions will be predicted by the HAPA variables of risk perception, self-efficacy, and outcome expectancies (H1)*,* and that the addition of the TPB variable subjective norms will increase the proportion of explained variance in intention (H2). In the volitional phase, the intention–behavior relationship is mediated by action control, as well as the planning variables of action planning and coping planning (H3). It is further hypothesized that experiences with AMR bacteria (H4) and experiences with foodborne infections (H5) lead to safer food-handling behaviors. To the best of our knowledge, this is the first study with a focus on the psychological process of safe food handling to mitigate the transmission risk of AMR bacteria via food.

## 2. Materials and Methods

### 2.1. Procedure

The study participants were recruited through an online panel of a commercial internet research company in January 2019 after the institutional ethics committee waived approval. The potential participants received an e-mail invitation to join a study about safe food handling. We defined quotas to obtain a sample representing the Swiss adult population according to age (38% between 18–38 years, 38% between 39–59 years, and 24% between 60–100 years) and gender. Only people who occasionally prepared food themselves were eligible to participate.

First, all participants were provided with a brief introduction to the study, including a short explanation about the study procedure. The introduction emphasized the anonymity of the data collection and processing procedures. After this instruction, the respondents were asked to provide their informed consent to participate in the study. Only those who consented proceeded to answer the questions for assessing the inclusion criterion, that is, regular food preparation. The respondents who fulfilled the inclusion criterion were asked to read a short text about AMR. The participants received a small remuneration from the internet research company if they answered all questions. Completion of the survey took approximately 13 min.

### 2.2. Survey

The objective of the questionnaire was to assess the psychological determinants that may influence safe food-handling behaviors with regards to AMR. The survey was designed to include components of the HAPA and the construct subjective norms of the TPB. Items were adapted from previous studies and translated into German [20,37,50,58]. Specific food-handling practices regarding chilling, cooking, cleaning, and separation were addressed in the questions.

The following definition of AMR was presented to the participants at the start of the online survey: “Antibiotic resistance means that antibiotic medicines no longer defeat bacteria. Among humans and animals carrying resistant bacteria, an antibiotic treatment therefore does not result in killing or growth inhibition of the bacteria. The affected humans and animals remain ill for a longer time. Livestock animals, and therefore, meat and meat products, may contain antibiotic-resistant bacteria. Vegetables and fruit might be exposed to antibiotic-resistant bacteria through the environment (e.g., through groundwater).”

The questionnaire was programmed in Questback [59]. Seven-point Likert scales were used to assess all items. Self-reported food preparation behavior was measured with 10 items covering chilling, cleaning, cooking, and separation safety practices (response scale: 1 = never to 7 = always). A series of seven items investigated participants’ risk perception of unsafe food handling (response scale: 1 = unlikely to 7 = very likely). Intention, subjective norms, and outcome expectancy (response scale: 1 = do not agree at all to 7 = fully agree) regarding safe food handling were measured with two items. Positive and negative outcome expectancies of implementing safe food-handling measures were assessed with one item. In addition, we assessed self-efficacy (four items, response scale: 1 = do not agree at all to 7 = fully agree), action planning (four items, response scale: 1 = do not agree at all to 7 = fully agree), coping planning (three items, response scale: 1 = do not agree at all to 7 = fully agree), and action control (two items, response scale: 1 = does not apply at all to 7 = fully applies). All items and their descriptive statistics can be found in the Appendix A, Table A2.

At the end of the questionnaire, all participants were asked questions about their level of education, household composition, and subjective health condition (response scale: 1 = very bad to 7 = very good). Furthermore, participants were asked whether they had experienced ineffective antibiotic treatments and gastrointestinal infections caused by food.

### 2.3. Sample

The participants were adult members of an online survey panel operated by an internet marketing and research company, and all were from the German-speaking part of Switzerland. The final study sample consisted of 665 food preparers. The mean age of the 336 (50.5%) female and 329 (49.5%) male participants was 47.91 years (SD = 15.90), with an age range of 18–80 years. The majority of the participants (n = 314, 47.2%) reported that they prepare food one to three times a week, 268 (40.3%) four to six times a week, and 83 (12.5%) more than six times a week. Meat and fish were consumed by 582 (87.5%) of the participants, while 46 (6.9%) participants reported that they consume meat but not fish, 4 (0.6%) participants reported that they consume fish but not meat, and six (0.9%) participants reported that they do not consume meat or fish.

Almost 16% of the participants had experienced a situation in which an antibiotic treatment was not effective (n = 104), and 12% had been affected by a gastrointestinal infection resulting from the consumption of home-cooked food (n = 81). A few participants (n = 44, 6.6%) could not answer these two questions. Three participants (0.5%) reported having a very bad health condition while 117 (17.6%) felt very healthy (M = 5.61, SD = 1.11). The majority of the participants (n = 258, 38.8%) lived with a partner, 170 (25.6%) in a single-person household, 148 (22.3%) with children, and 62 (9.4%) in another household composition. The majority of the participants (n = 320, 48.1%) had completed vocational school, 118 (17.7%) had completed higher secondary school, and 158 (23.8%) had a university degree. A few of the participants reported low levels of education as either primary school (n = 4, 0.6%) or obligatory levels of secondary school (n = 38, 5.7%) as their highest level of education.

### 2.4. Analyses

Data analyses were conducted using SPSS 16 [60]. The data were cleaned as follows. Some people dropped out after reading the definition of AMR (n = 18). Respondents who completed the survey within the 5th percentile (<4.58 min) of the time required by all respondents to complete the survey were excluded (n = 32), as were respondents with a mean value over all items of 1 or 7, before recoding inverse items (n = 10). Given that the respondents had the option to suspend the survey at any time and to proceed later, no maximum duration was set. Twenty-seven (4%) participants of the final study sample (N = 665) did not complete the survey, which resulted in gaps in the demographic data collected at the end of the survey.

All reverse-scaled items were reverse coded. As a preliminary data analysis step, the internal consistency of each construct was tested (see Appendix A, Table A2). Items were removed if removing them substantially improved Cronbach’s alpha. Then, an average score was calculated over all items belonging to the same construct. The average scores were used in all further analyses. Owing to the low internal consistency of the outcome expectancy items, we selected one variable that best represented perceived positive outcome expectancy and one variable that best represented perceived negative outcome expectancy to be included as single items in the following analyses.

Descriptive statistics were reported for the scales. Then, a multiple hierarchical regression analysis was conducted to test the motivational phase with intention as the dependent variable and risk perception, positive and negative outcome expectancies, self-efficacy, and subjective norms as the independent variables. Self-efficacy, risk perception and outcome expectancy were included as predictors of intention in step 1 (H1). Subjective norms were added in step 2 (H2). A second hierarchical regression analysis was conducted to test the relationships between the measured theoretical determinants and safe food-handling behavior as the dependent variable. Self-efficacy, risk perception, and outcome expectancy were included in step 1; subjective norms in step 2; intention in step 3; and action planning, coping planning, and action control in step 4. Demographic variables were not included in the regression analyses because they are not part of the underlying theories.

Multiple mediation analyses were then performed to test whether planning (action planning and coping planning) and action control mediated the association between intention and behavior (see Figure 1). The mediation analysis was conducted with the multiple mediation macro PROCESS in SPSS, using model 4 [61]. The bootstrapping method was employed with 10,000 samples to assess the indirect effects between intention and behavior [62]. To test the effect of experiences with AMR or food poisoning on preventive food-handling behaviors, a Welch’s t-test for independent samples was conducted [63]. Statistical significance was set at 5% in all analyses.

## 3. Results

### 3.1. Descriptive Results

Table 1 presents the means, standard deviations, and correlations of the study variables. In general, the participants had a high level of action planning, self-efficacy, and intention. Risk perception and negative outcome expectancy had the lowest mean scores, where a low score on negative outcome expectancy implied that few disadvantages were perceived when implementing safe food-handling measures.

Correlations among the variables showed that intention was positively related to risk perception, self-efficacy, positive outcome expectancy, and subjective norms (Table 1). As expected, a negative relationship emerged between intention and negative outcome expectancy. Safe food handling was positively associated with action planning, coping planning, action control, and self-efficacy. All correlations between the predictors and safe food-handling behaviors were in the predicted direction. The predictor intention exhibited the strongest correlation with safe food handling (r = 0.71, *p* = 0.001).

### 3.2. Determinants of Behavioral Intention

The fit of the extended HAPA model was analyzed sequentially. First, we tested the motivational phase with a hierarchical linear regression analysis on intention. This revealed that risk perception, positive outcome expectancy, and self-efficacy contributed significantly to the regression model and accounted for 56% of the variation in intention. Incorporation of subjective norms in the regression model did not increase the explained variance in intention significantly, ∆F(1634) = 3.50, *p* = 0.06. All regression statistics are reported in Table 2. The intention to adopt safe food-handling practices was significantly related to risk perception, positive outcome expectancies, and most significantly to self-efficacy. No significant relationship was detected between intention and negative outcome expectancies.

### 3.3. Determinants of Safe Food Handling

First, a hierarchical regression analysis was performed to assess the predictive values of all proposed theoretical determinants on safe food-handling behaviors (Table 3). Positive outcome expectancy and self-efficacy contributed significantly to the model. The addition of subjective norms to the regression model significantly explained an additional 2% of the variation in safe food-handling behaviors, ∆F(1634) = 25.91, *p* = 0.001. After the inclusion of intention, positive outcome expectancy was no longer a significant predictor of behavior, ∆F(1633) =114.92, *p* = 0.001. The inclusion of action planning, coping planning, and action control in the fourth step explained an additional 2% of the variation in safe food-handling behaviors, where coping planning and action control were the only additional significant predictors, ∆F(3630) = 14.13, *p* = 0.001.

A mediation analysis was performed to study the explanatory power of volitional variables over and above the effect of behavioral intentions. Unstandardized regression coefficients of indirect effects were considered significant when zero was not included in the 95% confidence interval (95%CI) around the estimated coefficient of the relationship between intention and behavior or between a mediator (action planning, coping planning, or action control) and behavior [61]. The hypothesized model suggested potential indirect effects of intention through action planning, coping planning, and action control on safe food handling (Figure 1). These volitional variables were conceptualized as mediators and intention as a predictor of safe food-handling behaviors (H3).

The results are presented in Figure 2 and highlight a partial mediation of the intention–behavior relationship by other factors: there were significant indirect effects of intention on safe food handling, through coping planning (B = 0.06, SE = 0.01, 95%CI [0.04–0.08]) and through action control (B = 0.03, SE = 0.01, 95%CI [0.01–0.06]). We did not find a significant indirect effect of intention through action planning on safe food handling (B = 0.01, SE = 0.01, 95%CI [-0.00, 0.02]). Positive and significant paths were found directed from intention to action planning, coping planning, and action control (see Figure 2). Coping planning and action control predicted behavior significantly, but action planning did not. Intention still had a significant direct effect on behavior after including the three mediators, although the direct effect was weaker than before including the mediators.

### 3.4. Negative Experiences

To test the effects of negative experiences with AMR or food poisoning on preventive food-handling behaviors, we compared the food-handling behaviors of participants with and without negative AMR experiences and those of the participants with and without food poisoning experiences. Participants who had negative AMR experiences (M = 5.12, SD = 1.09, n = 104) implemented a greater number of safe food-handling behaviors than the participants who had no negative AMR experiences (M = 4.86, SD = 1.06, n = 534), t(143.59) = 2.17, *p* = 0.03 (H4). The effect size d = 0.24 corresponds to a small effect of AMR experiences [64]. By contrast, participants who had experience with food poisoning (M = 5.04, SD = 1.07, n = 81) did not implement a greater number of safe food-handling behaviors than the participants who had no experience with food poisoning (M = 4.88, SD = 1.06, n = 57), t(104.60) = 1.22, *p* = 0.23 (H5).

Additionally, intention was significantly stronger among the participants with AMR experience (M = 5.79, SD = 1.48, n = 104) than among the participants without AMR experience (M = 5.32, SD = 1.64, n = 534), t(156.08) = 2.92, *p* = 0.004, d = 0.30. The difference between the intentions of the participants with food poisoning experience and those without food poisoning experience was not statistically significant, t(111.76) = 0.74, *p* = 0.46. Neither experiences with AMR, t(133.99) = 1.86, *p* = 0.07, nor experiences with food poisoning, t(101.51) = 1.72, *p* = 0.09, significantly increased risk perception.

## 4. Discussion

In this study, we demonstrated that the motivational phase of the HAPA model can explain food preparers’ food-handling behaviors to reduce the transmission risk of AMR. Higher levels of risk perception, positive outcome expectancies, and self-efficacy predicted stronger intention to adopt safe food-handling behaviors; self-efficacy was the strongest predictor of intention. The data suggested that as food preparers become more confident in their belief that they can implement safe food-handling practices, the intention for safe food handling becomes stronger, even when faced with difficulties. Moreover, our results revealed an association between risk perception and intention, which indicated that the riskier a person perceives the negative consequences of unsafe food handling, the stronger is the motivation to act.

Contrary to assumptions, both positive and negative outcome expectancies showed little or no relation to intention. Although stronger beliefs about the antibacterial and protective effects of safe food-handling practices—that is, positive outcome expectancies—strengthened intention, positive outcome expectancies had the lowest impact on intention compared to self-efficacy and risk perception. The fact that the positive outcome is not associated with noticeable individual advantages might explain the low impact of this factor. Assessing the positive consequences of risky food handling compared to positive consequences of safe food handling might offer more predictive utility in predicting food preparers’ intention. Negative outcome expectancies were rated as low among our sample, which indicates that the food preparers did not associate safe food-handling behaviors with negative attributes (see Table 1). However, we should treat these findings with caution because the beliefs about negative and positive outcomes were measured using one item each only. In the case of negative outcome expectancies, the item addressed the time investment for implementing safe food-handling behaviors. Food preparers might anticipate other negative outcomes than the extra time effort for implementing safe food-handling behaviors, for example, killing health-promoting bacteria or weakening the immune system [65]. Furthermore, the relationship between negative outcome expectancies and intentions seems to depend on the nature of the behavior. In previous studies, for instance, no significant relationship between negative outcome expectancies and intention was found for physical activity [66], whereas negative outcome expectancies predicted internet addiction behavior [67]. Thus, some behaviors may result in a more favorable balance of positive over negative outcome expectancies, which in turn, influence intention. Construal level theory may provide an appropriate explanation for this phenomenon: the importance of low-level aspects (such as cons) depends more on the value of the high-level aspects (such as pros) than vice versa [68,69]. Given that our sample valued the expected positive outcomes of safe food preparation as not extremely high (see Table 1), this might explain our finding that negative outcome expectancies were not related to intention, because the subjective importance of cons depends more on the existence of pros than the importance of pros depends on the existence of cons.

Contrary to the TPB and previous research, subjective norms were not found to be associated with the intention for safe food handling. This implies that development of the intention to adopt safe food-handling behaviors does not depend on the expectations of others and the motivation to comply with these expectations. This is inconsistent with the results of a study in which subjective norms explained a part of the variance in the intention to implement safe food-handling behaviors [37]. The public’s general lack of awareness about AMR bacteria in food products might explain our findings. If those people important to a food preparer are unaware of the risks of AMR, social pressure to implement safe food-handling behaviors will be absent. Alternatively, the private context of food handling might explain why the norms of others do not affect intention. The motivation to fulfil others’ behavioral norms may be weak if the behavior is not observable and, therefore, not evaluable. Although subjective norms were not a significant predictor of intention, they constituted one of the direct predictors of safe food-handling behaviors. This finding indicated that there were normative influences regarding safe food handling in our sample, and they changed behaviors rather immediately, and not indirectly, based on someone’s intentions. Thus, we can assume that safe food-handling behaviors are determined by the expectations of important others if these people are present during food preparation, for example, when hosting guests for dinner or cooking with a friend.

In terms of the predictors of behavior, our findings partially supported the volitional phase as outlined in the HAPA model [36]. It appears that in terms of safe food-handling behaviors, personal intention has the strongest influence on future behaviors compared to planning and controlling. We examined whether these findings support the more traditional social cognition models such as the TPB. Because intention and pre-intentional determinants accounted for 58% of the variance in safe food-handling behaviors in our study, additional factors are needed to explain the relationship between intention and behavior. Our findings might imply that the participants were in the motivational rather than the volitional phase (see HAPA, [70]). The results further suggest that self-efficacy not only strengthens behavior through intentions but also predicts behavior directly. This is in line with the more specific HAPA models (compared to our hypothesized model), which claim that perceived self-efficacy plays a crucial role in both phases (e.g., [71]).

Coping planning and action control mediated the relationship between intentions and behavior, although the effects were rather weak. These findings highlighted that anticipating difficulties (e.g., time pressure) that might hinder safe food handling can improve behavioral enactment. Food preparers who are aware about possible barriers can consider in advance means to overcome them. Furthermore, we found that monitoring current food-handling behaviors and comparing them to the intended behaviors enhanced goal achievement. This control process seemed to help the food preparers by not acting according to habitual behavior patterns. A weak effect of action control might have occurred because it was not an intended behavior change due to an intervention. Action control might be more important during the initial weeks after a planned change (i.e., an intervention) [45].

Moreover, in contrast to previous research conducted using the HAPA, action planning did not mediate the relationship between intention and behavior. In these studies, specifying plans to perform the intended action facilitated health behaviors, such as breast self-examination or physical exercise [34,49]. Our results suggest that food preparers who are motivated to adopt safe food-handling behaviors plan these in advance but do not implement them in practice. It is likely that these behaviors were not implemented because food handling is a habitual behavior, given that it has been implemented in many previous meals, and there is no conscious implementing involved. However, because other habitual behaviors, such as seat belt use, have been successfully predicted in previous studies using the HAPA, a few habitual behaviors seem to rely on action planning [72]. A similar result regarding action planning as a non-significant mediator between intention and safe food-handling behavior was reported by Mullan et al. [29], although their study did not include coping planning and action control as additional mediators. Our results indicate that it is important to additionally address the two strategies of coping planning and action control to improve safe food handling. Because our data could not confirm the hypothesized dual mediation of action planning and coping planning, we believe that a serial mediation model might predict safe food-handling behaviors. Thus, intention predicts safe food handling with a causal chain linking the two planning mediators; intention influences action planning, which in turn determines coping planning, which in turn influences safe food-handling behavior. Such a model was found to well predict the performance of physical exercise [40].

In addition to a few psychosocial determinants, our data demonstrated the influence of negative experiences on present behavior. While the participants with AMR experiences implemented a greater number of safe food-handling behaviors than the participants without negative AMR experiences, food poisoning experiences had no significant effect on the implementation of safe food-handling behavior. Presumably, the consequences of AMR are perceived as more dangerous and life-threatening than those of food poisoning. This finding indicates that people try to avoid the same negative experiences in the future. An alternative explanation might be that food poisoning is not attributed to one’s own kitchen behaviors. Therefore, it is not perceived as controllable through one’s own actions. Unlike other research carried out in this area [55], we did not find a significant difference in risk perception between participants with and without negative AMR experiences. Thus, AMR experiences may not determine intention through risk perception. Additional data should be collected to determine exactly how AMR experiences affect intention.

Moreover, it was surprising that a greater number of participants reported experiences with AMR compared to those with food poisoning. The number of AMR bacterial infections reported in Switzerland is considerably lower than the number of foodborne infections [3,13]. However, it must be noted that some of these foodborne infections may have been caused by resistant bacteria.

### Strengths and Limitations

This study has several strengths, such as the inclusion of a representative sample from the German-speaking part of Switzerland and the application of a theoretical framework to explain safe food-handling behaviors for mitigating AMR transmissions. Anonymous survey completion by the participants promotes the accuracy and truthfulness of the data. Furthermore, this work was able to fill a research gap because the determinants of safe food-handling behaviors to mitigate AMR transmission had not been investigated thus far. Although the preventive measures are the same for bacteria with and without an AMR gene responsible for foodborne illnesses, this study demonstrated that the underlying factors that foster the implementation of these measures partly differ. This is not surprising because individual health consequences, social impact, and public awareness of foodborne AMR bacteria and foodborne bacteria without AMR differ. Safe food handling is just as effective against the foodborne transmission of AMR bacteria as it is against the foodborne transmission of non-resistant bacteria. Preventing the spread of AMR bacteria through food is even more urgent than preventing the spread of non-resistant bacteria because in AMR bacteria, horizontal gene transfer additionally accelerates the spread of AMR, e.g., [73].

A limitation of our study is the self-reporting of safe food-handling behaviors, which might be biased owing to social desirability [74], albeit previous studies have not found any relationship between social desirability and self-reported health behaviors (e.g., a study about physical health behaviors [75]). Another potential limitation involves the retrospective assessment strategy used in our cross-sectional study. Different measurement points were recommended to test the model assumptions because the motivational and volitional phases are temporally sequential [70]. As our study did not include planning or action control as an intervention strategy, all variables were measured at the same time point. Moreover, there are limitations related to generalizability beyond Switzerland [76]. The scope of this study was constrained to food preparers in Switzerland and AMR awareness might differ across cultures and countries.

## 5. Conclusions

The findings of our survey indicate that interventions that promote safe food handling to reduce the transmission risk of AMR should primarily consider enhancing food preparers’ self-efficacy, because doing so strengthens their intention to implement these practices. Moreover, food preparers would benefit from tangible food-handling recommendations to increase their perceived positive outcome expectancies. It is important to provide risk information and communicate the possible consequences of an infection with AMR bacteria. Risk information enhances food preparers’ awareness about the risks of AMR, which, in turn, strengthens their intention to act. Interventions should further advice food preparers about how to overcome barriers to the implementation of safe food-handling behaviors (e.g., habits or lack of time). This strategy should be mainly aimed at food preparers who are already motivated to implement safe food-handling behaviors. Therefore, it is important to identify their perceived barriers to preparing food in a safe manner.

More research is needed to assess whether different safe food-handling behaviors are governed by different psychosocial factors. Washing hands after contact with raw meat, for example, may be perceived as a subjective norm, but thorough cooking of raw meat (e.g., beef) may not be. Moreover, further investigations are needed to test communication strategies that target the relevant determinants. This study, thus, serves as a base for future intervention studies aiming to mitigate the transmission of AMR bacteria among consumers through food products.

## Figures and Tables

**Figure 1 ijerph-17-02546-f001:**
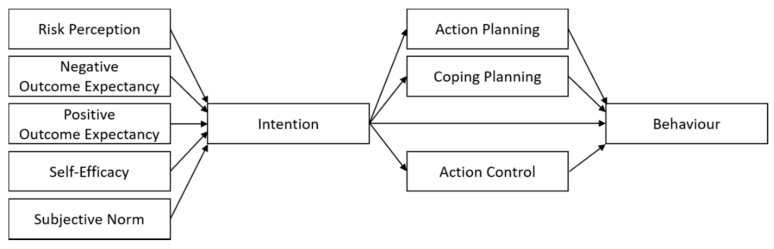
Hypothesized model of the relationships between the variables in the motivational phase and the volitional phase according to health action process approach (HAPA), extended with the theory of planned behavior (TPB) subjective norms.

**Figure 2 ijerph-17-02546-f002:**
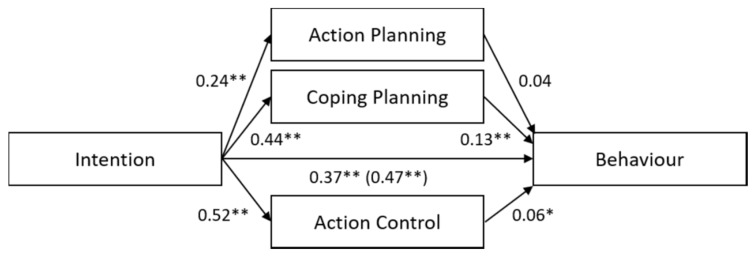
Unstandardized regression coefficients (B) are reported for the relationships between intention and safe food-handling behavior mediated by action planning, coping planning, and action control. The unstandardized regression coefficients between intention and behavior before controlling for the mediators are given in parentheses. * *p* < 0.05, ** *p* < 0.01.

**Table 1 ijerph-17-02546-t001:** Descriptive statistics and correlation matrix of all variables.

Variable	M	SD	1		2		3		4		5		6		7		8		9		10	
1. Behavior	4.89	1.06	-																			
2. Intention	5.38	1.62	0.71	**	-																	
3. Self-efficacy	5.65	1.26	0.69	**	0.74	**	-															
4. Risk perception	3.92	1.32	0.25	**	0.30	**	0.26	**	-													
5. Positive outcome expectancy	4.89	1.70	0.30	**	0.30	**	0.30	**	0.25	**	-											
6. Negative outcome expectancy	1.92	1.49	−0.16	**	−0.21	**	−0.24	**	0.03		0.04		-									
7. Subjective norms	4.83	1.76	0.45	**	0.39	**	0.45	**	0.22	**	0.25	**	−0.06		-							
8. Action planning	5.86	1.11	0.35	**	0.34	**	0.41	**	0.01		0.06		−0.39	**	0.19	**	-					
9. Coping planning	4.52	1.71	0.53	**	0.42	**	0.53	**	0.22	**	0.22	**	−0.09	*	0.46	**	0.39	**	-			
10. Action control	4.52	1.94	0.51	**	0.44	**	0.49	**	0.25	**	0.25	**	−0.10	*	0.55	**	0.31	**	0.64	**	-	
11. Age	47.91	15.90	0.20	**	0.10	*	0.17	**	0.01		0.03		0.18	**	0.10	*	0.21	**	0.21	**	0.19	**
12. Education			0.26		0.08	*	0.06		0.07		−0.01		−0.07		0.02		0.11	**	0.06		0.03	
13. Gender ^#^			0.06		0.10	*	0.10	*	0.05		−0.01		−0.18	**	0.03		0.14	**	−0.02		0.03	

*Note.* ** *p* < 0.01, * *p* < 0.05, ^#^ 1 = female, 0 = male.

**Table 2 ijerph-17-02546-t002:** Summary of hierarchical regression analysis of variables predicting intention.

Variable	Step 1	Step 2
	β	β
Risk perception	0.11 ***	0.10 ***
Positive outcome expectancy	0.07 *	0.07 *
’Negative outcome expectancy	−0.05	−0.05
Self-efficacy	0.68 ***	0.65 ***
Subjective norms		0.06
R^2^	0.56	0.56
F (df_1_, df_2_)	203.55 *** (4, 635)	164.18 *** (5, 634)

*Note.* Adjusted R^2^ is reported. *** *p* < 0.001, ** p* < 0.05.

**Table 3 ijerph-17-02546-t003:** Summary of hierarchical regression analysis of variables predicting behavior.

Variable	Step 1	Step 2	Step 3	Step 4
	β	β	β	β
Risk perception	0.06	0.04	0.00	−0.01
Positive outcome expectancy	0.09 **	0.07 *	0.05	0.04
Negative outcome expectancy	−0.01	−0.01	0.01	0.02
Self-efficacy	0.64 ***	0.58 ***	0.31 ***	0.22 ***
Subjective norms		0.16 ***	0.14 ***	0.07 *
Intention			0.42 ***	0.41 ***
Action planning				0.04
Coping planning				0.15 ***
Action control				0.07*
R^2^	0.48	0.50	0.58	0.60
F (df_1_, df_2_)	150.49 *** (4, 635)	130.30 *** (5, 634)	147.25 *** (6, 633)	108.99 *** (9, 630)

*Note.* Adjusted R^2^ is reported. *** *p* < 0.001, *** p* < 0.01, ** p* < 0.05.

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
