# Peer review of "Understanding the Underlying Psychosocial Determinants of Safe Food Handling among Consumers to Mitigate the Transmission Risk of Antimicrobial-Resistant Bacteria"

_ijerph, 2020, doi:10.3390/ijerph17072546_

Round 1
Reviewer 1 Report
The manuscript is very well structured. With the increase in AMR rate different mitigation strategies are been taken into action,this work is one of that kind. Transmission of AMR through food chain is one of the most prominent route. Identifying the most basic factor to reduce AMR transmission i.e awareness of the food handler about risk of AMR is important.the only minor comment from my side: It will be good if the author can shorten the introduction part.
Reviewer 2 Report
Studies exploring the impact of the handling of food preparation at home sheds light on microbial prevention, as many incidents are associated with home versus outside meal preparation. The authors utilization of statistical analysis to examine HAPA from a motivational perspective is a unique and important study. The authors provide clear discussion about the signficance and limitations of the study, why strengthen the manuscript. There was some confusion surrounding the explanation concerning the regression coefficients of the relationship (figure 2).
Also, the authors indicate, "The hierarchical regression revealed that risk
perception, positive outcome expectancy, and self-efficacy contributed significantly to the regression...", positive outcome expectancy appeared to have the lowest impact, when compared to the 2 other factors. Indeed table 2 would indicate it would be the list factor to impact intention. Additionally, the authors indicate positive outcome expectancy impact is not very high (line 397).
The inclusion of the impact of positive and negative outcomes when purchasing the mobile device may not be the most effective comparison tool. Many individuals will change economic expenditures based on cons. The factors that influence perspectives for food handling are very different from those associated with buying power.
Reviewer 3 Report
This manuscript has an appropriate structure, the methods are described in depth and the results are discussed correctly.
However, it requires a slight review of English to eliminate certain topological errors.
It is also necessary to include a list of acronyms, and it would be beneficial to have a graphical abstract to better eplain the methodology, and to avoid getting lost among so much statistical explanation.
